# Using the *PEAR1* Polymorphisms Rs12041331 and Rs2768759 as Potential Predictive Markers of 90-Day Bleeding Events in the Context of Minor Strokes and Transient Ischemic Attack

**DOI:** 10.3390/brainsci13101404

**Published:** 2023-09-30

**Authors:** Yanjie Xu, Dongxiao Yao, Weiqi Chen, Hongyi Yan, Dexiu Zhao, Lingling Jiang, Yicong Wang, Xingquan Zhao, Liping Liu, Yongjun Wang, Yuesong Pan, Yilong Wang

**Affiliations:** 1Department of Neurology, Beijing Tiantan Hospital, Capital Medical University, Beijing 100069, China; christine19840602@163.com (Y.X.); wangyicongccmu@126.com (Y.W.);; 2China National Clinical Research Center for Neurological Diseases, Beijing 100070, China; 3Advanced Innovation Center for Human Brain Protection, Capital Medical University, Beijing 100070, China; 4Beijing Key Laboratory of Translational Medicine for Cerebrovascular Disease, Beijing 100050, China; 5Department of Neurology, Beijing Long Fu Hospital, Beijing 100010, China; 6Department of Neurology, Aviation General Hospital, Beijing 100025, China; zhaodexiu1983@163.com

**Keywords:** *PEAR1* gene, platelet aggregation, dual antiplatelet therapy, stroke, transient ischemic attack, genetic polymorphism

## Abstract

In this study, we explored the relationship between the platelet endothelial aggregation receptor 1 (*PEAR1*) polymorphisms, platelet reactivity, and clinical outcomes in patients with minor stroke or transient ischemic attack (TIA). Randomized controlled trial subgroups were assessed, wherein patients received dual antiplatelet therapy for at least 21 days. Platelet reactivity was measured at different time intervals. Genotypes were categorized as wild-type, mutant heterozygous, and mutant homozygous. Clinical outcomes were evaluated after 90 days. The rs12041331 polymorphism predominantly influenced adenosine diphosphate channel platelet activity, with the AA genotype displaying significantly lower residual platelet activity to the P2Y12 response unit (*p* < 0.01). This effect was more evident after 7 days of dual antiplatelet treatment (*p* = 0.016). Mutant A allele carriers had decreased rates of recurrent stroke and complex endpoint events but were more prone to bleeding (*p* = 0.015). The rs2768759 polymorphism majorly impacted arachidonic acid (AA) channel platelet activity, which was particularly noticeable in the C allele carriers. Our regression analysis demonstrated that rs12041331 AA + GA and rs2768759 CA predicted 90-day post-stroke bleeding. In conclusion, the *PEAR1* polymorphisms rs12041331 and rs2768759 interfere with platelet aggregation and the performance of antiplatelet drugs. These genetic variations may contribute to bleeding events associated with minor stroke and TIA.

## 1. Introduction

The combination of aspirin with either clopidogrel or ticagrelor for 21 days is the standard therapy for minor stroke or Transient Ischemic Attack (TIA) [1,2,3]. A known challenge is high platelet reactivity (HPR), a phenomenon where platelet aggregation is not effectively suppressed by standardized medication, with an incidence rate of 5–65% [4,5]. The VerifyNow assay is commonly used to monitor platelet responses to antiplatelet drugs [6]. Currently, the causes of platelet hyperreactivity are uncertain, with factors such as age, sex, smoking habits, medication compliance, drug dosage, and genetic polymorphisms garnering significant attention. Numerous genes have been examined for their reactivity to aspirin, either alone or in concert with clopidogrel or ticagrelor [7,8,9,10]. Platelet Endothelial Aggregation Receptor 1 (*PEAR1*) is a platelet transmembrane protein that promotes platelet adhesion and aggregation, plays a role in thrombosis, and helps maintain platelet aggregation homeostasis [11]. Some studies have found a significant correlation between this gene locus and agonist-induced platelet aggregation activity [12]. This connection influences platelet reactivity to various agonists [13]. Since the COX1/thromboxane A2 pathway is strongly inhibited by aspirin, maximum aggregation relies on other secondary signaling pathways. Genetic variation in *PEAR1* is thus considered a critical determinant of residual platelet function during aspirin treatment and might be a crucial determinant of residual platelet function during antiplatelet therapy [11,14], thereby affecting the clinical outcomes of stroke and TIA patients. This study aims to evaluate the effectiveness of aspirin in conjunction with clopidogrel or ticagrelor for treating minor strokes or TIA. We will investigate the association between *PEAR1* gene polymorphisms, namely rs12041331, rs2768759, and rs56260937, platelet reactivity (as detected by VerifyNow), and the 90-day clinical outcome. The goal is to guide patients in taking effective measures to reduce platelet hyperreactivity, decrease the risk of bleeding, and improve clinical efficacy.

## 2. Materials and Methods

### 2.1. Study Population

A pre-designated subgroup analysis was conducted for the PRINCE trial, a randomized, multicenter, prospective, active-control, open-label trial with blinded endpoints. The rationale and design of the PRINCE trial have been previously outlined [15]. In brief, the trial randomly assigned patients with acute minor ischemic stroke (with an NHISS score ≤ 3) or moderate- to high-risk TIA (those with an ABCD2 score ≥4 or ≥50% neck or intracranial vascular stenosis explaining their clinical symptoms) to receive either a combination of ticagrelor and aspirin or clopidogrel and aspirin within 24 h of symptom onset. The patients were administered aspirin (300 mg on the first day and 100 mg daily from days 2–21) along with either ticagrelor (180 mg on the first day and 90 mg twice daily from days 2–90) or clopidogrel (300 mg on the first day and 75 mg daily from days 2–90). Patient enrollment commenced in August 2015, and the follow-up period concluded in June 2017. The PRINCE trial, which included 675 patients at 26 centers, is registered under the number NCT02506140 on ClinicalTrials.gov, accessed on 24 July 2020.

### 2.2. Platelet Reactivity Measurement

Patients were assessed at baseline and at 7 ± 2 days and 90 ± 7 days post-enrollment using the VerifyNow test. This test measured the Aspirin Reaction Unit (ARU) for aspirin and the P2Y12 Response Unit (PRU) for clopidogrel or ticagrelor. Platelet function tests were performed following the standardized procedure manual by qualified personnel at each study center. These personnel were blinded to the treatment assignments. While patients and investigators were aware of the study drug allocation, the data regarding platelet reactivity remained unavailable until the completion of the trial. To ensure the accuracy and reproducibility of these methods, we conducted two separate training sessions for all examiners at each study center.

### 2.3. PEAR1 Genotyping

Blood samples were delivered to Beijing Tiantan Hospital using cold chain logistics and subsequently stored at -80℃. We evaluated three single-nucleotide polymorphisms (SNPs) of the *PEAR1* gene: rs12041331, rs2768759, and rs56260937. Genotyping was conducted using the Sequenom MassArray iPLEX platform (Sequenom, San Diego, CA, USA). In cases where the initial results were unclear, Sanger sequencing was performed using an ABI 3500 Genetic Analyzer (Applied Biosystems, Foster City, CA, USA). The call rate for each SNP exceeded 98.5%.

### 2.4. Outcome

The effectiveness of the treatment was assessed based on the incidence of stroke (ischemic or hemorrhagic) or composite vascular events (hemorrhagic or ischemic stroke, transient ischemic attack, myocardial infarction, or vascular death) within a 90-day period. Safety outcomes were identified according to the PLATO standard for bleeding events after 90 days [16].

### 2.5. Statistical Analysis

Continuous variables are presented as mean standard deviations, medians are displayed with interquartile ranges, and categorical variables are denoted as percentages. Student’s *t*-test or the Wilcoxon test were employed for the analysis of continuous variables, while the χ^2^ test was used for the categorical variables. Comparisons of baseline characteristics were made between the gene test group and the group excluded from the gene test, as well as comparisons regarding the incidence of major bleeding events. We evaluated the ARU and PRU values of different genotypes and compared the incidence of clinical endpoint events between the wild-type and mutant strains. Cox proportional hazards regression was used to assess differences in the incidence of clinical outcomes, and these were reported as 95% confidence interval (CI) hazard ratios (HR) over the 90-day follow-up period. To determine whether specific genotypic classes had distinct therapeutic effects, we examined the interaction effects of genotypic therapy. The χ^2^ test was utilized to verify the Hardy–Weinberg equilibrium. Statistical significance was established at *p* < 0.05. All analyses were conducted using SAS 9.4 (SAS Institute, Cary, NC, USA).

## 3. Results

### 3.1. Baseline Characteristics of Study Participants

Out of the 675 patients enrolled in the PRINCE trial, this subgroup analysis included 372 patients (55.1%). The baseline characteristics for both the included and excluded patient subgroups are presented in Table 1. Both groups showed a well-balanced distribution of baseline data, with no significant differences in age, sex, blood pressure, body mass index, antiplatelet drugs, concurrent diseases, smoking history, drug combination, randomization time, NHISS score, ABCD2 score, SSS-TOAST classification, creatinine clearance, blood glucose levels, or platelet count (*p* > 0.05). However, the groups differed significantly in terms of pulse rate and serum creatinine levels (*p* < 0.05). The subgroup participants had a median age of 61 years, with 311 (83.6%) patients experiencing minor stroke and 61 (16.4%) having TIA (Table 1).

Table 1 also delineates the baseline characteristics of patients who experienced a bleeding endpoint event (66 patients) and those who did not (302 patients). Both groups had an average age of 60.6 years and were primarily male. The mean systolic blood pressure at enrollment was between 151 and 155 mmHg. The distribution of medication history and previous disease history was well balanced between the two groups at the time of enrollment. Notably, the risk of bleeding in the ticagrelor/aspirin subgroup was higher than in the clopidogrel/aspirin group (*p* < 0.05), and the risk of bleeding in patients with minor stroke was higher than in those with TIA (*p* < 0.05).

**Table 1 brainsci-13-01404-t001:** Baseline characteristics and 90-day bleeding events of genetic subgroups in the PRINCE trial.

Characteristic	Genetic Subgroups in the PRINCE Trial		*p* Value	90-Day Bleeding Events in Genetic Subgroups		*p* Value
	Included in the Genetic Subgroup (*n* = 372)	Excluded from the Genetic Subgroup (*n* = 303)		No-90-Day Bleeding Events (*n* = 302)	90-Day Bleeding Events (*n* = 66)	
Age (years)						
Mean (standard deviation)	60.6 (9.0)	61.0 (8.4)	0.481	60.6 (9.1)	60.6 (8.5)	0.983
Median (interquartile range)	61.0 (55.0–67.0)	62 (54.0–67.0)		61 (55–67)	59 (53–67)	
Female sex	93(25.0)	88 (29.0)	0.238	73 (23.9)	20 (30.3)	0.273
Systolic blood pressure (mmHg)						
Mean (standard deviation)	154.2 (22.3)	152.9 (21.3)	0.427	154.8 (22.5)	151.5 (21.6)	0.271
Median (interquartile range)	152 (140–170)	151 (140–168)		154.5 (140–170)	150.5 (140–160)	
Diastolic blood pressure (mm Hg)						
Mean (standard deviation)	88.8 (13.2)	88.2 (12.6)	0.519	89.0 (13.2)	88.2 (13.4)	0.669
Median (interquartile range)	88.5 (80–97)	87.0 (80–95)		89 (80–98)	87 (80–96)	
Body mass index *						
Mean (standard deviation)	25.1 (3.9)	24.8 (3.7)	0.306	25.3 (3.9)	24.5 (3.6)	0.157
Median (interquartile range)	25.0 (22.8–27.3)	24.2 (22.5–27.3)		25.0 (22.9–27.2)	24.2 (21.5–27.5)	
Pulse rate (beat/min; mean (SD))	76.8 (10.8)	74.3 (10.8)	0.003	77.0 (10.7)	75.7 (11.2)	0.379
Antiplatelet therapy, *n* (%)						
Clopidogrel + aspirin	185 (49.7)	151 (49.8)	0.979	165 (53.9)	22 (33.3)	0.002
Ticagrelor + aspirin	187 (50.3)	152 (50.2)		141 (46.1)	44 (66.7)	
Medical history, n (%)						
Hypertension	233 (62.6)	178 (58.8)	0.303	189 (61.8)	44 (66.7)	0.455
Dyslipidemia	25(6.7)	16(5.3)	0.436	21(6.9)	4(6.0)	0.813
Diabetes mellitus/Hyperglycemia (baseline > 11.1 mmol/L)	92 (24.7)	72 (23.7)	0.790	78 (25.5)	14 (21.2)	0.465
Ischemic stroke	68 (18.3)	53 (17.5)	0.791	58 (19.0)	10 (15.2)	0.469
Transient ischemic attack	12 (3.2)	6 (2.00)	0.318	10 (3.3)	2 (3.0)	0.921
Coronary artery disease	25 (6.7)	26 (8.6)	0.363	19 (6.2)	6 (9.1)	0.396
Ex-smoker	32 (8.6)	19 (6.3)		28 (9.2)	4 (6.1)	
Drug use before randomization (No (%))						
Proton pump inhibitor	2 (0.5)	3 (1.0)	0.495	1 (0.3)	1 (1.5)	0.231
Statin	33 (8.9)	33 (10.9)	0.380	27 (8.8)	6 (9.1)	0.945
Aspirin	88 (23.7)	58 (19.1)	0.157	69 (22.6)	19 (28.8)	0.279
Clopidogrel	7 (1.9)	8 (2.6)	0.506	6 (2.0)	1 (1.5)	0.809
Ticagrelor	0 (0.0)	0 (0.0)	--	0 (0.0)	0 (0.0)	--
Anticoagulants	0 (0.0)	0 (0.0)	--	0 (0.0)	0 (0.0)	--
Defibrase	1 (0.3)	0 (0.0)	0.366	1 (0.33)	0 (0.0)	0.642
Time to randomization after onset of symptoms (h; median (IQR))	13.9(6.6)	14.4(6.8)	0.366	13.8(6.6)	14.9 (6.7)	0.220
Time to randomization after onset of symptoms (No (%))						
<12 h	214 (57.5)	178 (58.8)	0.750	132 (43.1)	26 (39.4)	0.577
≥12 h	158 (42.5)	125 (41.3)		174 (56.9)	40 (60.6)	
Qualifying event (No (%))						
Minor stroke	311 (83.6)	253 (83.5)	0.971	250 (81.7)	61 (92.4)	0.033
Transient ischemic attack	61 (16.4)	50 (16.5)		56 (18.3)	5 (7.6)	
Baseline NIHSS score (median (IQR))				1.6 (1.1)	1.9 (1.0)	0.074
Baseline ABCD2 (median (IQR))	4.85 (0.9)	4.62 (0.8)	0.177	4.9 (1.0)	4.8 (0.4)	0.898
SSS-TOAST stroke subtype (No (%))						
Large artery atherosclerosis	169 (54.3)	135 (53.4)	0.999	133 (53.2)	36 (59.0)	0.908
Non-Large artery atherosclerosis	142 (45.7)	118 (46.6)		117 (46.8)	25 (41.0)	
Serum creatinine, μmol/L	72.7 (18.8)	68.9 (18.7)	0.009	73.4 (18.7)	69.4 (18.9)	0.114
Plasma glucose, mmol/L	6.2 (2.2)	6.2 (2.4)	0.854	6.3 (2.2)	6.5 (2.4)	0.988
platelet count	220.4 (57.6)	217.3 (61.4)	0.501	220.2 (57.9)	221 (56.9)	0.914

* Data are represented as either the median with its interquartile range or as a count (percentage). NIHSS refers to the National Institute of Health Stroke Scale; SSS-TOAST stroke subtype refers to the Stroke Subtype as per the Stop Stroke Study Trial of Org 10172 in Acute Stroke Treatment Stroke Etiology Classification).

### 3.2. PEAR1 Gene Test Results

Individual variants were eliminated if the genotyping rate was less than 95%. The distribution of polymorphisms for rs12041331, rs2768759, and rs56260937 complied with the Hardy–Weinberg equilibrium (*p* > 0.05), suggesting that this patient population exhibited a certain degree of population representativeness (Table 2).

### 3.3. Relationship between Platelet Aggregation Pre- and Post-Treatment and Various SNPs

Carriers of the rs12041331 AA genotype displayed significantly lower platelet residual activity compared to the GG and GA genotypes at baseline as measured by PRU (*p* < 0.01). This difference became even more pronounced after seven days of dual antiplatelet drug treatment (*p* = 0.021). Notably, carriers of the A allele demonstrated higher platelet inhibition rates and significant differences in residual activity. These A allele carriers also exhibited greater platelet reactivity to ADP. A similar trend was observed for ARU, although the differences were not statistically significant. For rs2768759, the AA genotype was associated with significantly lower levels of platelet residual activity compared to the CA genotype (*p* = 0.033). Additionally, the platelet inhibition rate of ARU in the AA group was significantly higher than that in the CA group (*p* = 0.032). This suggests that the platelet reactivity of the C allele carriers to ARU was significantly greater. No substantial correlation was observed between the rs56260937 polymorphism and ARU or PRU levels. However, the rs12041331 and rs2768759 polymorphisms had notable impacts on platelet function and platelet aggregation and influenced the activity of antiplatelet drugs (refer to Figure 1).

### 3.4. Relationship between Various SNPs and Clinical Outcomes

Patients were categorized into groups based on the mutations in two single-nucleotide polymorphisms (SNPs): those carrying the wild-type allele (wild-type homozygote) and those carrying mutant alleles (mutant heterozygote and mutant homozygote). We then compared the occurrence of endpoint events between these groups. For both SNPs, there was no statistically significant difference in the incidence of endpoint events between patients with or without mutations (Table 3). However, those carrying the rs12041331 mutant allele demonstrated lower incidences of recurrent stroke, ischemic stroke, and multiple key events. Further, those carrying the ‘a’ allele at rs12041331 were more likely to experience bleeding events (*p* = 0.0146). Similarly, those carrying the ‘a’ allele at rs2768759 exhibited lower rates of recurrent stroke, ischemic stroke, and multiple key events. Additionally, carriers of the ‘a’ allele at rs12041331 showed a higher likelihood of bleeding events (*p* = 0.0112).

### 3.5. Logistic Regression Analysis

A logistic regression analysis was conducted to identify the risk factors for bleeding events. The findings indicated a positive correlation between bleeding events and the following independent risk factors: the antiplatelet treatment strategy employed by the ticagrelor and aspirin groups, the onset of minor stroke as an entry event, the AA + AG genotype at the rs12041331 locus, and the CA genotype at the rs2768759 locus (Figure 2). These factors were each independently associated with an increased risk of bleeding events.

## 4. Discussion

A subgroup analysis of the PRINCE trial confirmed that polymorphisms of the *PEAR1* gene, specifically SNPs rs12041331 and rs2768759, influence platelet function, aggregation, and the effectiveness of antiplatelet drugs. There was a statistically significant difference in bleeding event incidence between patients with and without these mutations, with carriers of the rs12041331 and rs2768759 alleles experiencing an elevated risk of bleeding.

*PEAR1* gene variants have been linked to platelet aggregation induced by various agonists [17,18] and the responses to several antiplatelet drugs, including aspirin [10,17,19], clopidogrel [20], and ticagrelor [21]. Since the COX1/thromboxane A2 pathway is strongly inhibited by aspirin, maximum aggregation relies on other secondary signaling pathways. Genetic variation in *PEAR1* is thus considered a critical determinant of residual platelet function during aspirin treatment [11]. Notably, the intron variant rs12041331 in *PEAR1* is a significant genetic modifier of platelet reactivity. Polymorphisms in rs12041331 can cause variations in gene and protein expression in patients treated with aspirin [22,23,24]. The *PEAR1* rs12041331 A allele has been significantly linked with reduced agonist-induced platelet aggregation in both African Americans and Europeans, accounting for approximately 15% of total phenotypic variation in platelet reactivity [18,25]. Our study found that platelets with different genotypes exhibited varying levels of baseline adenosine diphosphate (ADP) activity, and significant variations in ADP activity persisted even after dual antiplatelet intervention. The VerifyNow test corroborated that the *PEAR1* rs12041331 polymorphism primarily affects the ADP channel. This finding aligns with a previous turbidimetric test, indicating that rs12041331 in *PEAR1* impacts ADP-induced aggregation, much like arachidonic acid (AA). However, there was no significant correlation between AA and AA [26]. Unlike the GG and GA genotypes, the AA homozygotes of the intron rs12041331 of *PEAR1* influence ADP-induced aggregation [27,28,29]. The rs12041331 polymorphism, resulting in a G to A substitution, has been associated with reduced *PEAR1* expression [28]. Faraday et al. found that the major G allele is independently linked with higher platelet aggregation irrespective of aspirin therapy and that it does not alter the relationship between PRU and rs12041331 [30]. Moreover, the AA genotype of rs12041331 has been associated with an increased response to ticagrelor in healthy individuals [21]. Research on SNP rs2768759 is relatively sparse. Some studies have shown that the C allele of rs2768759, which responds to different agonists in healthy individuals, is linked to increased AA- and ADP-induced platelet aggregation. This association is stronger after antiplatelet drug administration, suggesting that the C allele is linked to natural platelet aggregation and reduced responsiveness to antiplatelet drugs [10]. The AA variant of rs2768759 was associated with lower platelet aggregation than the CA variant, but no significant difference was observed in AA- and ADP-induced platelet aggregation between these two genotypes [10,31]. Our study revealed that the gene polymorphism rs2768759 significantly differs from the baseline ARU and ARU differences, primarily affecting the AA channel, unlike rs12041331. C allele carriers had low ARU levels and high platelet inhibition rates after seven days of treatment. No significant differences were observed among the rs56260937 genotypes. However, the greater the difference in the TT genotype, the stronger the antiplatelet effect and the higher the bleeding risk. Consistent with previous studies [32], Kim et al. analyzed the effect of *PEAR1* variants on platelet function by sequencing the 1Q21.1 region of the chromosomes in 104 subjects. Platelet function was assessed using a PAP-4 Aggregometer following stimulation with collagen, ADP, or epinephrine. The results showed that rs56260937 was significantly linked with increased platelet aggregation across all three agonist phenotypes.

Research regarding the association between rs12041331, a potent genetic determinant of *PEAR1*, and cardiovascular outcomes under antiplatelet therapy is sparse, and the literature contains inconsistent results [33,34,35,36]. In Lewis’s study, carriers of the rs12041331 A allele exhibited a significantly elevated risk of myocardial infarction compared to GG homozygotes among Caucasian subjects undergoing percutaneous coronary intervention. This factor was considered an independent risk factor for cardiovascular events and platelet response in patients receiving either aspirin alone or in combination with clopidogrel [35]. Conversely, a post hoc analysis of 13,547 participants in the ASPREE trial indicated no association between the genetic variant rs12041331 and cardiovascular events in a healthy elderly population of European descent taking low-dose aspirin, with no clinically significant bleeding being observed [36]. Additionally, a meta-analysis encompassing six studies with 9914 patients revealed a link between the A allele of rs12041331, the AA homozygote of rs2768759, and an increased risk of ischemic events in patients with coronary heart disease receiving aspirin and P2Y12 receptor inhibitors [34]. Moreover, carrying the small T allele of rs56260937 independently predicted revascularization events in patients undergoing percutaneous coronary intervention for acute myocardial infarction [37]. A study on bleeding events indicated that patients with the TT + CT genotype of *PEAR1* rs41273215 exhibited a three-fold higher incidence of significant bleeding and higher platelet inhibition rates in ACS and AF patients [38].

Research investigating the relationship between *PEAR1* and stroke is scant, and the results remain elusive. A single-center study in Shanghai suggested that the impact of the rs12041331 polymorphism on aspirin efficacy depended on the TOAST subtype. Patients with rs12041331 AA stroke with small-artery occlusion demonstrated the highest sensitivity to sole aspirin therapy, leading to favorable short-term functional outcomes [39]. Although this might appear inconsistent with cardiovascular research, it aligns with our study and the underlying biology. In our study, patients with the rs12041331 AA genotype experienced the most notable reduction in ARU or PRU following dual antiplatelet therapy, indicating the highest platelet activity; the rates of stroke recurrence, ischemic stroke, and complex endpoint events were lower, and bleeding events were more probable. Likewise, compared to the AA type, the CA type of rs2768759 showed a significant ARU decrease following antiplatelet therapy, strong drug responsiveness, and a notable statistical difference in bleeding events.

Our regression analysis revealed that the AA genotypes of rs12041331 and CA of rs2768759 were predictors of short-term bleeding events post-stroke. This study is the first to provide robust evidence regarding the outcomes of multiple SNPs and dual-antibody stroke therapy, especially with respect to their correlation with bleeding. However, the incidence of recurrent stroke and composite endpoint events was not statistically significant, warranting further research. Notably, some patients without CYP2C19 mutations still experience ischemic events. Aside from pharmacokinetic pathways, mutations in genes associated with platelet aggregation might also influence the effects of P2Y12 receptor inhibitors and aspirin.

The limited sample size of this study restricted a subgroup analysis of the potential impacts of different medication regimes. In addition to *PEAR1* gene mutations, some studies have shown significant associations between CYP2C19, P2RY12, and ABCB1 genotypes and ischemic clinical outcomes. However, due to the paucity of research, this study did not consider these factors. Future studies analyzing polygenetic scores in different ethnic populations may enhance our understanding of the relationship between *PEAR1* SNPs and clinical outcomes in stroke patients. Going forward, *PEAR1* should be further investigated as a critical factor for managing treatment in stroke patients. Genetic testing and assessments of the *PEAR1* polymorphism could guide platelet activity diagnosis and bleeding prediction and ultimately improve patient treatment processes.

## 5. Conclusions

The polymorphisms at the rs12041331 and rs2768759 sites of the *PEAR1* gene influence platelet aggregation and can interfere with the efficacy of antiplatelet drugs. These polymorphisms could be crucial determinants of bleeding events associated with dual antiplatelet therapy for minor stroke and TIA.

## Figures and Tables

**Figure 1 brainsci-13-01404-f001:**
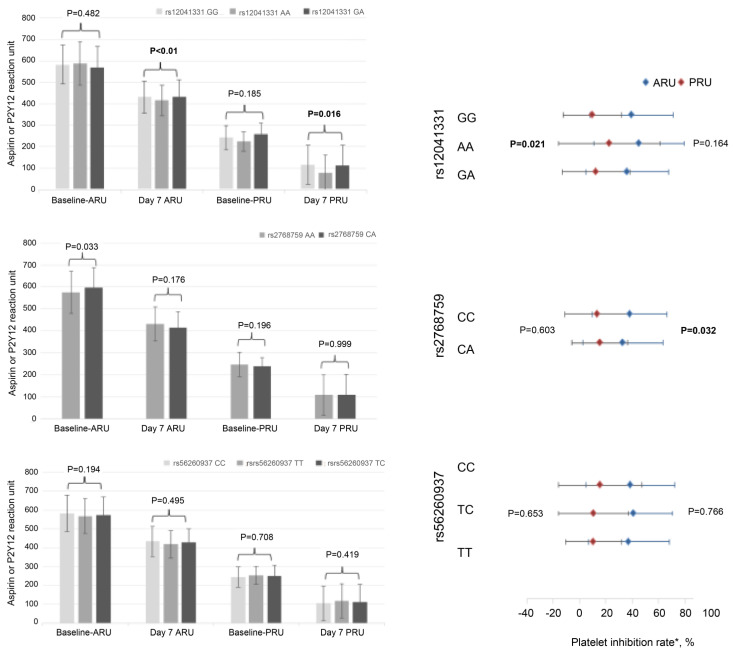
Analysis of platelet reactivity. The three genotypes of rs12041331 had different levels of PRU at baseline, day 7, and the platelet inhibition rates. For rs2768759, the platelet inhibition rate of ARU in the AA group was significantly higher than that in the CA group. ARU: Aspirin Reaction units; PRU: P2Y12 response unit. * Platelet Inhibition Rate: (baseline ARU/RRU−day 7 ARU/RRU) ÷ baseline ARU/RRU × 100.

**Figure 2 brainsci-13-01404-f002:**
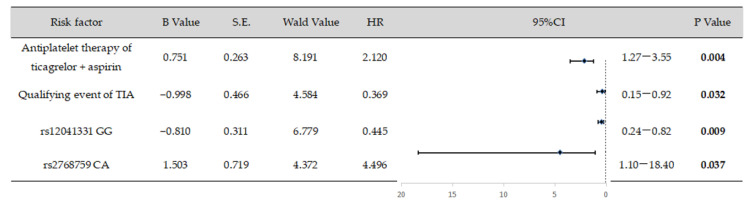
Logistic regression analysis of risk factors for bleeding events. Risk factors such as the ticagrelor and aspirin antiplatelet strategy, the initial event of mild stroke, and the presence of the AA + AG genotype at rs12041331 and the CA genotype at rs2768759 show a positive correlation with bleeding events. These factors are considered independent risk predictors for bleeding events.

**Table 2 brainsci-13-01404-t002:** Results of *PEAR1* gene analysis.

SNP Reference Number	Allele	Genotyping			Frequency of Allele Distribution	HWE *p*-Value
		Wild-Type Homozygote (No (%))	Heterozygous Mutation (No (%))	Homozygous Mutation (No (%))		
rs12041331	G/A	127	190	55	G 59.7% A 40.3%	0.732
rs2768759	A/C	324	44	0	A 94.02% C 5.78%	0.419
rs56260937	C/T	176	155	39	C 68.5% T 31.5%	0.666

HWE: Hardy–Weinberg equilibrium.

**Table 3 brainsci-13-01404-t003:** Analysis of clinical outcomes.

Outcomes	rs12041331 (*n* = 372)			rs2768759 (*n* = 368)			rs56260937 (*n* = 360)		
	GG (*n* = 127)	AA + AG (*n* = 245)	*p* Value	CA (*n* = 44)	AA (*n* = 324)	*p* Value	CC (*n* = 176)	CT + TT (*n* = 194)	*p* Value
Stroke	12 (9.45)	18 (7.35)	0.480	4 (9.09)	26 (8.02)	0.808	18 (10.23)	12 (6.19)	0.155
Composite events	12 (9.45)	19 (7.76)	0.575	4 (9.09)	27 (8.33)	0.865	19 (10.80)	12 (6.19)	0.110
Ischemic stroke	12 (9.45)	15 (6.12)	0.241	4 (9.09)	23 (7.10)	0.634	15 (8.52)	12 (6.19)	0.388
Major bleeding	0 (0.00)	3 (1.22)	0.554	0 (0.00)	3 (0.93)	1.000	3 (1.70)	0 (0.00)	0.107
Major or minor bleeding	0 (0.00)	3 (1.22)	0.554	0 (0.00)	3 (0.93)	1.000	3 (1.70)	0 (0.00)	0.107
Any bleeding	14 (11.02)	52 (21.22)	0.015	2 (4.55)	63 (19.44)	0.011	32 (18.18)	33 (17.01)	0.767

## Data Availability

The datasets used and/or analyzed during the current study are available from the corresponding author upon reasonable request.

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
