# Peer review of "Using the PEAR1 Polymorphisms Rs12041331 and Rs2768759 as Potential Predictive Markers of 90-Day Bleeding Events in the Context of Minor Strokes and Transient Ischemic Attack"

_brainsci, 2023, doi:10.3390/brainsci13101404_

Round 1
Reviewer 1 Report
Dear Authors
Congratulation to conducting a good study
Well written and designed, however comments need to be addressed
1. How many participants were included in the study, please mention in the study population section and also mention the study, which was used as a reference for the sample size calculation.
2. How did you randomized the cases?
3. Please improve in details the role of PEAR1 gene polymorphisms and platelet activity and how they affect the clinical outcome at 90 days!
4. Did you measure the clinical outcome at 30 days, If yes how much differences have been observed between 30 to 90 days?
5. Please explain the genotyping methodology in details and also explain the point of conducting the sanger sequencing, it seems incomplete sentence.
6. Explain the DNA isolation technique briefly.
7. Introduction need to be improved further, specially how the polymorphic variation affects the mechanism of action provided antiplatelet therapy in minoe stroke.
Can be improved
Author Response
Response to Reviewer 1
Comments and Suggestions for Authors
- How many participants were included in the study, please mention in the study population section and also mention the study, which was used as a reference for the sample size calculation.
Authors’ response: The PRINCE trial included 675 patients at 26 centers. This information was added to lines 76-77.
- How did you randomized the cases?
Authors’ response: Patients were allocated via a block randomization process by investigators at the clinical centers. The block randomization sequence was provided by an independent statistician using computer-generated random numbers with a block size of four. The block size was not listed in the Chinese version of the protocol (which was provided to the investigators) to prevent the investigators from speculating about the group assignment (DOI: 10.1136/bmj.l2211).
- Please improve in details the role of PEAR1 gene polymorphisms and platelet activity and how they affect the clinical outcome at 90 days!
Authors’ response: We have elaborated on this conclusion in the discussion. rs12041331 gene polymorphisms led to allele-specific differences in H3K4Me1 methylation, resulting in gene and protein expression differences and different platelet aggregation rates, especially after administration of antiplatelet drugs. AA type is highly reactive to platelets, has a lower proportion of recurrent stroke, ischemic stroke and complex endpoint events, and is also more prone to bleeding events. Similarly, compared with AA type, ARU decreased significantly after CA type antiplatelet treatment of rs2768759, and the rate of platelet inhibition was higher, with a statistically significant difference in bleeding events.
- Did you measure the clinical outcome at 30 days, If yes how much differences have been observed between 30 to 90 days?
Authors’ response: Unfortunately, this trial did not measure 30-day clinical outcomes.
5.Please explain the genotyping methodology in details and also explain the point of conducting the sanger sequencing, it seems incomplete sentence. Explain the DNA isolation technique briefly.
Authors’ response: Blood was collected in a container with ethylenediamine tetraacetic acid (EDTA). Genomic DNA was extracted from peripheral venous blood leukocytes using the standard salt-out method. SNP genotyping was performed using the SequenomMassARRAY system on an iPLEX SNP genotyping analyzer (Sequenom, USA). The primers were designed by online Analysis Design Suite software (Sequenom), and primers were synthesized by Invitrogen, USA. Based on the standard method provided by Sequenom MassARRAY SNP technique, PCR conditions, allele-specific primer elongation, and product analysis were performed. MassARRAY Typer 4.0 software (Sequenom) was used to process and analyze the generated mass spectrum to determine peak recognition and alleles. If the results were inconclusive, we used Sanger sequencing (ABI 3500 Genetic Analyzer, AppliedBiosystems). We have revised the sentence on lines 91-93.
- Introduction need to be improved further, specially how the polymorphic variation affects the mechanism of action provided antiplatelet therapy in minoe stroke.
Authors’ response: We've added it on lines 55-58.

Reviewer 2 Report
The title of the present manuscript is well formed, brief and informative, giving the direct aim of the paper. The abstract is well formed according to the requirements of the journal, giving the top highlights and conclusion from the whole work. The introduction can be more broad in aspect of other known or potential biomarkers. Nevertheless, its sounds intriguing as it is. The methods are discussed in detail, informative, noting the trial and its details. The whole methodology is well described in a comprehensive to the general reader manner as the chosen methods fit well to the given aims. No additional recommendations for the results section as it is strict, with good visualization of data. The discussion section is a bit larger than the average, but filled with useful and well given information on the topic and correlation of the results with those from other studies. A major advantage of the presented study is the use of a large cohort of patients, making the results more reliable, and additionally that the authors noted some limitations of their work. The conclusion relevantly gives the essence of the whole work.
Mild improvement of language might improve (only typographically) the present highly valuable work. This recommendation comes from very minor mistakes, which require grammatical and punctuational correction.
Author Response
Response to Reviewer 2
Comments and Suggestions for Authors
The title of the present manuscript is well formed, brief and informative, giving the direct aim of the paper. The abstract is well formed according to the requirements of the journal, giving the top highlights and conclusion from the whole work. The introduction can be more broad in aspect of other known or potential biomarkers. Nevertheless, its sounds intriguing as it is. The methods are discussed in detail, informative, noting the trial and its details. The whole methodology is well described in a comprehensive to the general reader manner as the chosen methods fit well to the given aims. No additional recommendations for the results section as it is strict, with good visualization of data. The discussion section is a bit larger than the average, but filled with useful and well given information on the topic and correlation of the results with those from other studies. A major advantage of the presented study is the use of a large cohort of patients, making the results more reliable, and additionally that the authors noted some limitations of their work. The conclusion relevantly gives the essence of the whole work.
Authors’ response: We thank the reviewer for the valuable comments.

Reviewer 3 Report
Dear Authors
PEAR1 Polymorphisms rs12041331 and rs2768759 as Potential Predictive Markers of 90-Day Bleeding Events in Minor Stroke or Transient Ischemic Attack, is a novel study with convincible background , methodology, results, and discussion However authors are requested to address the following recommendations
1. As one of the aim of the study is to evaluate the effectiveness of aspirin in conjunction with clopidogrel or ticagrelor for treating minor stroke or TIA. But, their effectiveness was not discussed and concluded in the manuscript
2.Authors are requested to specify selection criteria
3.Authors are requested to specify the allocation process in detail
Author Response
Response to Reviewer 3
Comments and Suggestions for Authors
PEAR1 Polymorphisms rs12041331 and rs2768759 as Potential Predictive Markers of 90-Day Bleeding Events in Minor Stroke or Transient Ischemic Attack, is a novel study with convincible background, methodology, results, and discussion However authors are requested to address the following recommendations
- As one of the aim of the study is to evaluate the effectiveness of aspirin in conjunction with clopidogrel or ticagrelor for treating minor stroke or TIA. But, their effectiveness was not discussed and concluded in the manuscript
Authors’ response: This manuscript focuses on the effects of genetic polymorphisms on platelet therapy for mild stroke and TIA, the effectiveness of which has been published in PRINCE's other articles and is therefore not discussed.
- Authors are requested to specify selection criteria
Authors’ response: From August 2015 to March 2017 in 26 study centers in China, the PRINCE trial enrolled patients aged 40-80 years who had experienced a minor acute ischemic stroke (National Institutes of Health Stroke Scale score of ≤3 at the time of randomization) or those with a moderate to high risk of transient ischemic attack (ABCD2 stroke risk score of ≥4 at the time of randomization or ≥50% stenosis of cervical or intracranial vessels that could account for the presentation) who could be treated with the study drug within 24 hours of symptom onset. Patients were excluded from the trial if they had a diagnosis of intracranial hemorrhage, acute coronary syndrome, or other pathology that could account for the neurological symptoms; had a modified Rankin scale score of more than 2 at randomization; or had a contraindication to ticagrelor, clopidogrel, or aspirin (See the Protocol,PMID: 28381198 and PMID: 31171523).
3.Authors are requested to specify the allocation process in detail
Authors’ response: Immediately after signing the written informed consent form, eligible patients were assigned to receive the following within one hour of randomization in a 1:1 ratio:
- Intervention (ticagrelor/aspirin): aspirin (a loading dose of 100-300 mg given as one to three 100 mg tablets on day 1, followed by 100 mg once daily until day 21) combined with ticagrelor (180 mg loading dose given as two 90 mg tablets on day 1, followed by 90 mg twice daily until day 90).
- Control (clopidogrel/aspirin): aspirin (a loading dose of 100-300 mg given as one to three 100 mg tablets on day 1, followed by 100 mg once daily until day 21) combined with clopidogrel (300 mg loading dose given as four 75 mg tablets on day 1, followed by 75 mg once daily until day 90).
Patients were allocated via a block randomization process by investigators at the clinical centers. The block randomization sequence was provided by an independent statistician using computer generated random numbers with a block size of four. The block size was not listed in the Chinese version of the protocol (which was provided to the investigators) to prevent the investigators from speculating about the group assignment. (See the Protocol,PMID: 28381198 and PMID: 31171523).

Reviewer 4 Report
The study aims to investigate the correlation between PEAR1 polymorphisms and platelet reactivity in patients with minor stroke or transient ischemic attack (TIA). The paper claims that certain PEAR1 polymorphisms, specifically rs12041331 and rs2768759, might serve as predictive markers for bleeding events following minor strokes or TIA. Overall, the manuscript organized, and the methodology seems rigorous. However, several areas require further clarification and substantiation.
Strengths:
The study's objective is clear, and the research question is of clinical relevance. The predefined subgroup analysis of a randomized controlled trial design strengthens the validity of the findings. Statistical methods seem appropriately used for data analysis. The manuscript is structured and easy to follow. Platelet reactivity was assessed with POC tests in 3 time-points, genetic SNP analyses were performed centrally. The study uses clinical adjudicated clinical end-points.
Areas for Improvement:
HOPR : The choice of this abbreviation is somewhat unconventional in the field, and it might lead to potential confusion among researchers and clinicians who are accustomed to the more generally used term "HPR" (High Platelet Reactivity).
Thoroughly revise the reference list to align with the journal’s citation style. This typically includes aspects like the order of authors, journal name abbreviations, year of publication, volume and issue numbers, and page numbers. Double-check each reference for accuracy, ensuring that all cited works are correctly represented and retrievable by readers. Confirm that all in-text citations correspond accurately to the reference list.
Table 1 presents the characteristics of the patients included and excluded from the current subgroup analysis. This as a supportive feature should be presented as Supplementary material. Table 1 should present patient characteristics according to the explored SNP subgroups.
The tables are poorly organized and hard to interpret. Especially, Table 3. Consider presenting these data – highlighting the most important ones in graphical form.
Line 279 please correct “Likewise, compared to the AA type, the CA type of rs2768759 showed a significant ARU decrease following antiplatelet therapy, strong drug responsiveness..., and a notable statistical difference in bleeding events.”
As this is a subgroup analysis, of a trial testing clopidogrel or ticagrelor based DAPT, results should concentrate on the data how the choice of P2Y12 inhibitor influences HPR and clinical results in patients with or without SNPs.
The discussion losts its focus and would profit from revision to improve readability. Specifically, despite that the two polymorphism had (rs12041331 and rs2768759) different impact on the platelet reactivity and clinical outcomes they are summarized together. Furthermore, the clinical impact of the findings remains poorly understood. Of note, secondary prevention and risk of bleeding events are influenced by the choice of the antiplatelets as well as by the use of monotherapy or DAPT. (see PMID: 34063551 and PMID: 34162232 ) A thorough discussion on this topic would greatly benefit the reader, particularly in understanding the decision-making process when it comes to selecting the appropriate therapeutic strategy for different types of ischemic strokes.
The paper offers valuable insights into the role of PEAR1 polymorphisms in predicting bleeding events in minor stroke and TIA patients. However, the above-mentioned areas require attention for the study to have a more robust scientific and clinical impact.
Author Response
Response to Reviewer 4
Comments and Suggestions for Authors
The study aims to investigate the correlation between PEAR1 polymorphisms and platelet reactivity in patients with minor stroke or transient ischemic attack (TIA). The paper claims that certain PEAR1 polymorphisms, specifically rs12041331 and rs2768759, might serve as predictive markers for bleeding events following minor strokes or TIA. Overall, the manuscript organized, and the methodology seems rigorous. However, several areas require further clarification and substantiation.
Strengths:
The study's objective is clear, and the research question is of clinical relevance. The predefined subgroup analysis of a randomized controlled trial design strengthens the validity of the findings. Statistical methods seem appropriately used for data analysis. The manuscript is structured and easy to follow. Platelet reactivity was assessed with POC tests in 3 time-points, genetic SNP analyses were performed centrally. The study uses clinical adjudicated clinical end-points.
Areas for Improvement:
HOPR : The choice of this abbreviation is somewhat unconventional in the field, and it might lead to potential confusion among researchers and clinicians who are accustomed to the more generally used term "HPR" (High Platelet Reactivity).
Authors’ response: We thank the reviewer for the suggestion, we have made the relevant change in line 43.
Thoroughly revise the reference list to align with the journal’s citation style. This typically includes aspects like the order of authors, journal name abbreviations, year of publication, volume and issue numbers, and page numbers. Double-check each reference for accuracy, ensuring that all cited works are correctly represented and retrievable by readers. Confirm that all in-text citations correspond accurately to the reference list.
Authors’ response: We have revised the reference list in accordance with the journal’s guidelines.
Table 1 presents the characteristics of the patients included and excluded from the current subgroup analysis. This as a supportive feature should be presented as Supplementary material. Table 1 should present patient characteristics according to the explored SNP subgroups.
The tables are poorly organized and hard to interpret. Especially, Table 3. Consider presenting these data – highlighting the most important ones in graphical form.
Authors’ response: We thank the reviewer for the suggestions. We have removed some content to highlight subgroup-associated patient characteristics, and converted Table 3 to Figure 1.
Line 279 please correct “Likewise, compared to the AA type, the CA type of rs2768759 showed a significant ARU decrease following antiplatelet therapy, strong drug responsiveness..., and a notable statistical difference in bleeding events.”
As this is a subgroup analysis, of a trial testing clopidogrel or ticagrelor based DAPT, results should concentrate on the data how the choice of P2Y12 inhibitor influences HPR and clinical results in patients with or without SNPs.
The discussion losts its focus and would profit from revision to improve readability. Specifically, despite that the two polymorphism had (rs12041331 and rs2768759) different impact on the platelet reactivity and clinical outcomes they are summarized together. Furthermore, the clinical impact of the findings remains poorly understood. Of note, secondary prevention and risk of bleeding events are influenced by the choice of the antiplatelets as well as by the use of monotherapy or DAPT. (see PMID: 34063551 and PMID: 34162232 ) A thorough discussion on this topic would greatly benefit the reader, particularly in understanding the decision-making process when it comes to selecting the appropriate therapeutic strategy for different types of ischemic strokes.
The paper offers valuable insights into the role of PEAR1 polymorphisms in predicting bleeding events in minor stroke and TIA patients. However, the above-mentioned areas require attention for the study to have a more robust scientific and clinical impact.
Authors’ response: We thank the reviewer for the valuable advice, which has helped broaden our knowledge. The selection of different P2Y12 inhibitors to discuss HPR and clinical outcomes has been published as PRINCE's main research direction. If the P2Y12 inhibitors are stratified, the current sample size makes it difficult to draw reliable conclusions regarding the SNP subgroups. We will continue to supplement the sample size in future studies and provide an in-depth discussion of these issues.
